# Ultra-sensitive polarization-resolved black phosphorus homojunction photodetector defined by ferroelectric domains

Shuaiqin Wu[1,2,6], Yan Chen[1,6], Xudong Wang[1✉], Hanxue Jiao[1,2], Qianru Zhao[1,2], Xinning Huang[1,2], Xiaochi Tai[1,2], Yong Zhou[1], Hao Chen[1], Xingjun Wang[1], Shenyang Huang[3], Hugen Yan[3], Tie Lin[1,2], Hong Shen[1,2], Weida Hu[1,2], Xiangjian Meng[1,2], Junhao Chu[1,2] & Jianlu Wang[1,4,5✉]

With the further miniaturization and integration of multi-dimensional optical information detection devices, polarization-sensitive photodetectors based on anisotropic low-dimension materials have attractive potential applications. However, the performance of these devices is restricted by intrinsic property of materials leading to a small polarization ratio of the detectors. Here, we construct a black phosphorus (BP) homojunction photodetector defined by ferroelectric domains with ultra-sensitive polarization photoresponse. With the modulation of ferroelectric field, the BP exhibits anisotropic dispersion changes, leading an increased photothermalelectric (PTE) current in the armchair (AC) direction. Moreover, the PN junction can promote the PTE current and accelerate carrier separation. As a result, the BP photodetector demonstrates an ultrahigh polarization ratio (PR) of 288 at 1450 nm incident light, a large photoresponsivity of 1.06 A/W, and a high detectivity of $1.27 \times 10^{11}$ cmHz$^{1/2}$W$^{-1}$ at room temperature. This work reveals the great potential of BP in future polarized light detection.

[1] State Key Laboratory of Infrared Physics, Shanghai Institute of Technical Physics, Chinese Academy of Sciences, No.500 Yutian Road, 200083 Shanghai, China. [2] University of Chinese Academy of Sciences, No.19 A Yuquan Road, 100049 Beijing, China. [3] State Key Laboratory of Surface Physics, Key Laboratory of Micro- and Nano-Photonic Structures (Ministry of Education), and Department of Physics, Fudan University, 200433 Shanghai, China. [4] Frontier Institute of Chip and System, Fudan University, 200433 Shanghai, China. [5] Shanghai Frontier Base of Intelligent Optoelectronics and Perception, Institute of Optoelectronics, Fudan University, 200433 Shanghai, China. [6]These authors contributed equally: Shuaiqin Wu, Yan Chen. ✉email: wxd0130@mail.sitp.ac.cn; jlwang@mail.sitp.ac.cn

Polarization is an important property of light, besides intensity, frequency, and phase. Correspondingly, polarization detectors play a vital role in environmental monitoring, remote sensing, medical detection, and navigation[1,2]. Polarization imaging technology is very efficient and reliable in identifying targets in a complicated environment. Traditional polarization detector technique usually requires the integration of complex optical structures to obtain and analyze the polarization information of targets, such as a polarization photodetector with a single quantum well sandwiched in plasmonic micro-cavity[3]. However, in order to satisfy the development of photodetectors in the direction of high integration and miniaturization, the use of polarization-sensitive materials as a functional layer is a more direct approach to realize polarization detection.

For the distinct lamellar structure and remarkable electrical and optical properties, two-dimensional (2D) materials are supposed important candidates for ultrathin devices and super integrated devices[4,5]. Among them, 2D materials with anisotropic structure and polarization sensitivity, such as BP, black arsenic (b-As), GeSe, GeAs, $ReS_2$, and $ReSe_2$, have shown great advantages and potential in the development of highly integrated polarization detectors[6,7]. To characterize the polarization sensitivity of the photodetectors, one crucial figure of merit PR is defined and given by the expression $PR = I_{ph\_maximum}/I_{ph\_minimum}$. In practical application, the PR of the photodetector is supposed to be higher than 20[8]. However, most polarization photodetectors based on anisotropic 2D materials show small PRs (commonly less than 10)[6]. Polarization imaging technology needs a polarization ratio as larger as possible. Therefore, it is urgent to enhance the PR of those 2D materials photodetectors by introducing physical or chemical methods. Among most anisotropic 2D materials, BP has a puckered hexagonal structure with armchair-shaped along armhair (AC) direction and zigzag-shaped along zigzag (ZZ) direction, and benefits from its thickness-tunable direct bandgap and ultrahigh carrier mobility of more than $1000\,cm^2V^{-1}s^{-1}$, making it one of the best candidates for polarization detection materials[9]. However, due to the low dichroic ratio and conductivity ratio, the polarization sensitivity of intrinsic BP-based polarization detectors is still unsatisfactory.

Several approaches to enhance the polarization ratio of photodetectors based on 2D materials have emerged in the past decade. The built-in electric field in PN junctions can greatly enhance the efficiency of linear dichroism photodetectors, enhancing the polarization ratio. Such as BP vertical homojunction (35)[10], $BP/MoS_2$ heterojunction (>100)[11], and graphene/$PdSe_2$/Ge heterojunction (112.2)[12]. Artificial metamaterials with sub-wavelength structures can spatially confine light on the nanometer scale and enhance light absorption of 2D materials. With bowtie apertures on BP surface, the polarization ratio of the detector increase to 8.7[13]. More interestingly, isotropic materials can realize polarization light detection if some non-centrosymmetric plasmonic nanoantennas are fabricated on the surface. For instance, the graphene mid-infrared photodetector with bulk photoresponse can realize calibration-free polarization detection at zero bias[14]. By tuning the orientation of nanoantennas, the polarization ratio of graphene photodetector even varies from positive to negative[8]. Another method is utilizing an external electric field to enhance the anisotropic PTE efficiency of BP, which is proposed by first-principles and model calculation[15]. Nevertheless, it is a lack of reports about real devices fabricated according to this method.

Considering that BP is very sensitive to the external electric fields of its highly susceptible electronic state, it brings many possibilities for tuning the band structure, electron transport, and photoelectric conversion process of BP. Under the modulation of a high electric field (1.9 V/nm), a single-layer BP becomes a Dirac semimetal with anisotropic dispersion, which is linear in the AC direction and quadratic in the ZZ direction[16]. By virtue of the Stark effect, the bandgap of a 5 nm-thick BP can be effectively reduced more than 0.18 eV by vertical electric field[17]. More than that, the photoluminescence (PL) peak of the 20-layer BP can be continuously tuned from 3.7 to 7.7 μm by a moderate displacement field up to 0.48 V/nm[18]. However, there are few studies on the electric field tuning BP anisotropic photoelectric properties, which are used to explore a photodetector with higher polarization sensitivity.

Ferroelectrics is a kind of dielectric material with spontaneous electric polarization. It provides a strong local field exceeding 1 V/nm, which can effectively improve the photoelectric properties of most semiconductor materials[19–21]. Here, we demonstrate an ultra-sensitive polarization-resolved photodetector with BP in-plane homojunction defined by ferroelectric domains. The dominant photoelectric conversion mechanism is photothermoelectric (PTE) effect of the device at zero or small bias, which converts heat into electricity, and enables ultra-broadband photodetection without cooling[22]. BP is supposed to be a kind of potentially promising thermoelectric material with anisotropic in-plane thermal conductivity, whose thermoelectric power can be improved via an external electric field[23]. Considering that the external electric field has an anisotropic tunable effect on the band structure of BP, the ultrahigh ferroelectric field enhances the thermoelectric conversion efficiency in the AC direction. Thus, the PTE current in the device is increased in the AC direction and causes an enhanced polarization detection ratio. More importantly, BP is p-doped and n-doped by the up-polarized ($P_{up}$) and down-polarized ($P_{down}$) ferroelectric domains to form an in-plane PN junction. Under illumination, the built-in field in the homojunction accelerates the separation of photo-generated electron-hole pairs and further improves the photocurrent collection efficiency. Therefore, based on this device architecture, combining PTE effect and photovoltaic (PV) effect, BP embraces an ultrahigh polarization sensitivity (PR = 288, at 1450 nm incident light), a large photoresponsivity of 1.06 A/W, and a high detectivity of $1.27 \times 10^{11}\,cm\,Hz^{1/2}W^{-1}$ at room temperature and atmospheric condition.

## Results

**Device structure designed based on photoelectric mechanism.** In order to obtain a low dark current and a high polarization ratio, and to combine the PTE effect and PV effect in one device, we designed and prepared a BP in-plane PN homojunction defined by the ferroelectric field. The few-layer BP is transferred on the $SiO_2/Si$ substrate, and the BP is doped into p-type and n-type with up-polarized and down-polarized ferroelectric domains, as shown in Fig. 1a. For traditional BP field-effect transistors (FET), the PTE photocurrent is affected by the electrical doping of the metal electrodes[24]. When the doping level is uniform, the carriers will be driven by the temperature gradient and transported in opposite directions, leading to a zero net current (Fig. 1b).

When introduced a PN junction in BP (BP in-plane PN junction), there will be a positive contribution to the PTE current on both sides (Fig. 1c). Such a mechanism worked well in graphene[24]. The thermoelectric conversion efficiency is governed by $ZT = S^2\sigma/(K_L + K_e)$, where $S$ is Seebeck coefficient, $\sigma$ is electrical conductivity, $K_L$ and $K_e$ are lattice and electronic components of the thermal conductivity, respectively[25]. The ZT value in the AC direction is certainly larger than that in the ZZ direction, which is consistent with the electronic band structure and carrier transport properties[26]. There are some methods to enhance ZT value, including modifying band structure, quantum

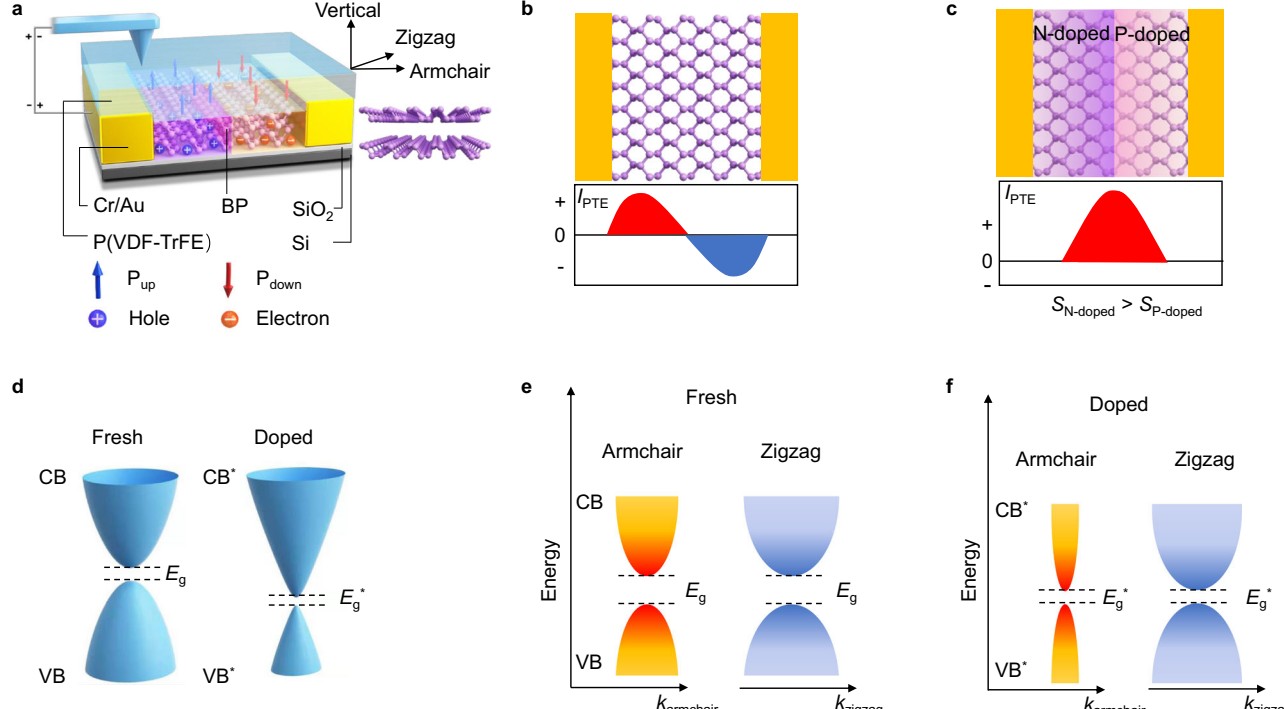

**Fig. 1 Device structure designed based on photoelectric mechanism. a** Structure schematic of BP in-plane PN homojunction defined by ferroelectric domains. $P_{up}$ and $P_{down}$ represent the ferroelectric copolymers are totally polarized upward and downward by piezoresponse force microscopy (PFM) conductivec probe. AC and ZZ axes correspond to the armchair and zigzag direction of BP crystal structure. **b** Photothermoelectric (PTE) current ($I_{PTE}$) in BP device at zero bias. Red (blue) area represents positive (negative) contribution to the PTE current. **c** PTE current in BP in-plane PN junction device at zero bias. $S_{N-doped}$ and $S_{P-doped}$ correspond to the Seebeck coefficient of n-doped BP and p-doped BP. **d** Band structure of BP in the fresh state and the electrical doped state. Under large electric field, bandgap decreases after electrical doping, and the band dispersion becomes nearly linear in the AC direction while remains parabolic in ZZ direction. **e** Anisotropic band structure of BP in the fresh state along the AC direction ($k_{armchair}$) and ZZ direction ($k_{zigzag}$). CB, VB, and $E_g$ correspond to conduction band, valence band and bandgap of BP in fresh state, respectively. **f** Anisotropic band structure of BP in the doped state along the armchair direction and zigzag direction. CB*, VB*, and $E_g^*$ correspond to conduction band, valence band, and bandgap of doped BP, respectively.

confinement effect, band convergence, and reducing lattice thermal conductivity[22]. The lattice thermal conductivity is almost determined by the intrinsic properties of the material, therefore electronic band engineering is the main strategy because $\sigma$ and $S$ are dominated by the band structure and Fermi level. Here, the Seebeck coefficient is tuned by the external electric field. The Seebeck coefficient of the n-doped area is larger than that of p-doped area for BP[23], as shown in Fig. 1c, which also promotes the PTE current flow directionally from the n-doped to the p-doped area.

Moreover, the electric field has a remarkable tunable effect on the band structure in the AC direction, while the band structure in the ZZ direction hardly changes[16]. Figure 1d shows the band structure of BP in the fresh state and doped state. With an external electric field, the bandgap ($E_g$) is narrowed owing to the giant Stark effect[16,17]. Figure 1e and f is cross-sectional views (in the armchair and zigzag direction) of the two states in Fig. 1d. Angle-resolved photoemission spectroscopy (ARPES) has proven that high electric fields can change the band structure of BP anisotropically[16]. Under the doping of the electrostatic field, the shape of the band structure in the AC direction changes obviously, while that in the ZZ direction is almost unchanged. It is not difficult to speculate that the electrical conductivity of the AC direction has increased even more. Thus, the ZT in the AC direction is larger than that in the ZZ direction, leading to a higher PTE current in the AC direction. Consequently, the designed BP in-plane PN homojunction device will have more sensitive in polarization resolution.

**Raman spectroscopy evolution of BP under ferroelectric field.** Generally, Raman spectroscopy can provide structural and electronic information without destructing the material[27]. More importantly, the Raman spectroscopy technique is also an efficient method to measure the thermal conductivity and explore the electron-phonon interaction of 2D materials[26,28]. The classical Raman scattering theory denotes the Raman intensity ($I$) is given by:

$$I \sim |\mathbf{e}_i \times \mathbf{R} \times \mathbf{e}_s|^2 \tag{1}$$

where $\mathbf{e}_i$ and $\mathbf{e}_s$ are the incidents and scattered light, respectively, $\mathbf{R}$ is the Raman tensor, which is related to the derivative of the dielectric tensor for the phonon vibration[29]. For the polarized Raman spectroscopy of BP, the Raman intensity reflects the anisotropic electron-photon and electron-phonon interactions[27]. Therefore, by comparing the Raman spectral intensities of excitation light with different polarization angles, the coupling interaction of electron-photon and electron-phonon can be obtained. According to our previous analysis and inference, when BP is doped by an external electric field, the electronic structure in the AC direction will change obviously, but the electronic structure in the ZZ direction will not change. As a result, as shown in Fig. 2a, the $A_g^2$ phonons along the AC direction should perform a remarkable strong resonance enhancement at 0°.

As shown in Fig. 2b, three typical Raman peaks $A_g^1$ (out of plane mode), $B_{2g}$ (in-plane mode along ZZ direction), and $A_g^2$ (in-plane mode along AC direction) of BP are located at 366, 443, and 472 cm$^{-1}$, respectively. It was observed and proved that the

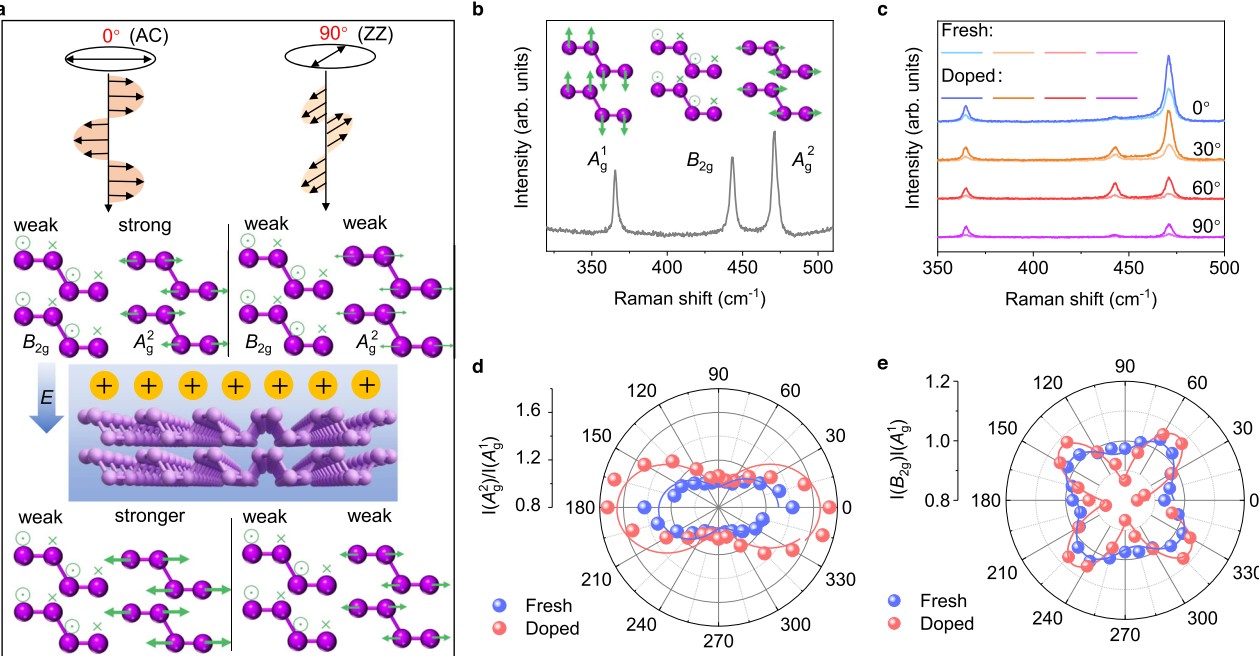

**Fig. 2 Electrical tunable Raman scattering spectra of BP. a** Schematic illustration of electrical tunable anisotropic Raman vibration modes in BP. The ferroelectric field enhances the $A_g^2$ vibration under 0° polarization light. The 0° (90°) light is linearly polarized along AC (ZZ) direction. The green arrows represent the vibration directions of raman modes. The $E$ represents the direction of ferroelectric field and the yellow spheres represent the p-type doping effect on BP by ferroelectric field. **b** Typical measured Raman spectrum of exfoliated BP flake. The insets illustrate three Raman vibration modes of BP. **c** Polarized Raman spectra of BP sample with different polarization angles (0°, 30°, 60°, 90°). **d, e** The measured (dots) and fitted (line) angular-resolved Raman intensity ratio of the $A_g^2/A_g^1$ mode and $B_{2g}/A_g^1$ mode. The solid lines are fitted using the function of $I(A_g) = (a\cos^2\theta + c\sin^2\theta)^2$ and $I(B_{2g}) = e^2\sin^2 2\theta$, where **a**, **c**, and **e** represent the magnitude of three independent components of Raman tensor, $\theta$ is the angle between the incident polarization and the AC direction.

intensity of these three peaks is different from each other, as a result of the anisotropic structure of BP[29]. To further study the electron-photon and electron-phonon interactions of BP, polarized Raman spectra were measured on the same sample. Because the anisotropic electrical and optical properties are mainly decided by intralayer structure for few-layer BP, we mainly discuss the in-plane Raman peaks of BP. To verify the regulating effect of the electric field on BP, we performed the polarized Raman spectra characterization when BP was in the fresh state and doped state (n-doped by a ferroelectric field). To control variables and ensure the reliability of the data, we performed this experiment under the same conditions (measured at the same position on the same sample, using a 514 nm laser with the same power of 1.25 mW and the same integration time) in step of 15° from 0° to 180°.

The experiment results are shown in Fig. 2c–e, and the detailed data can be found in Supplementary Fig. 1. Figure 2c shows the Raman spectra of different polarization angles (0°, 30°, 60°, and 90°). The intensities of the three Raman peaks are all enhanced after electrostatic doping. As mentioned above, the Raman intensity reflects the electron-photon and electron-phonon coupling interactions. From the near-infrared absorption spectrum (Supplementary Fig. 2), the electron-photon interaction seems nearly unchanged. Thus, it is reasonable to infer that the ferroelectric field enhances the electron-phonon coupling interaction of BP. By comparing the ratio of in-plane and out-of-plane phonon vibrations, we can estimate the degree of molecular deformation and quantify the electron-phonon coupling interactions[30]. Figure 2d and e show the extracting intensity ratio of $A_g^2/A_g^1$ and $B_{2g}/A_g^1$, respectively. When the polarization of the excitation laser is aligned along the AC direction, the intensity ratio of $A_g^2/A_g^1$ reaches the maximum value, which is

inconsistent with the previous reports[17]. After the modulation of the ferroelectric field, the ratio of $A_g^2/A_g^1$ increased from 1.42 to 1.73 at 0° but did not change at 90°. In contrast, the ratio of $B_{2g}/A_g^1$ is 0.93 (0°) and 0.97 (90°) in the fresh state, and decreases to 0.84 and 0.87 after doping. The BP crystal shows more obvious deformation in the AC direction than that in the ZZ direction. It can be concluded that $A_{2g}$ along the AC direction shows stronger electron-phonon interaction with polarization light at 0° while $B_{2g}$ in zigzag direction shows weaker interaction for polarization light at 0° and 90°. The electron-phonon interaction plays an important role in reducing thermal conductivity[31,32]. Therefore, the enhanced electron-phonon interaction in the AC direction indicates that the ZT in the AC direction decreases while ZT in the ZZ direction doesn't change. In other words, the ZT ratio is improved by the ferroelectric field. The $P_{up}$ and $P_{down}$ ferroelectric fields have similar effects on BP according to experimental results shown in Supplementary Fig. 3.

**Optoelectronic properties of the BP in-plane homojunction.** Based on the above analysis and experiment results, we designed and prepared a BP in-plane homojunction defined by ferroelectric domains. Poly(vinylidene fluoride-trifluoroethylene) (P(VDF-TrFE)) is employed as the ferroelectric layer. Arbitrary patterned ferroelectric domains can be obtained through piezoelectric force microscopy (PFM) technology. The device structure is shown in Fig. 3a, and the device can usually work at zero bias for low dark current ($I_{dark}$)[20]. Cross-section of the device analyzed by high-resolution transmission electron microscopy (HR-TEM), as shown in Fig. 3b. There is a very clear interface between BP and P(VDF-TrFE) and the BP thickness is 4.8 nm (~9 layers). Energy-dispersive X-ray spectroscopy (EDS) elemental mapping of

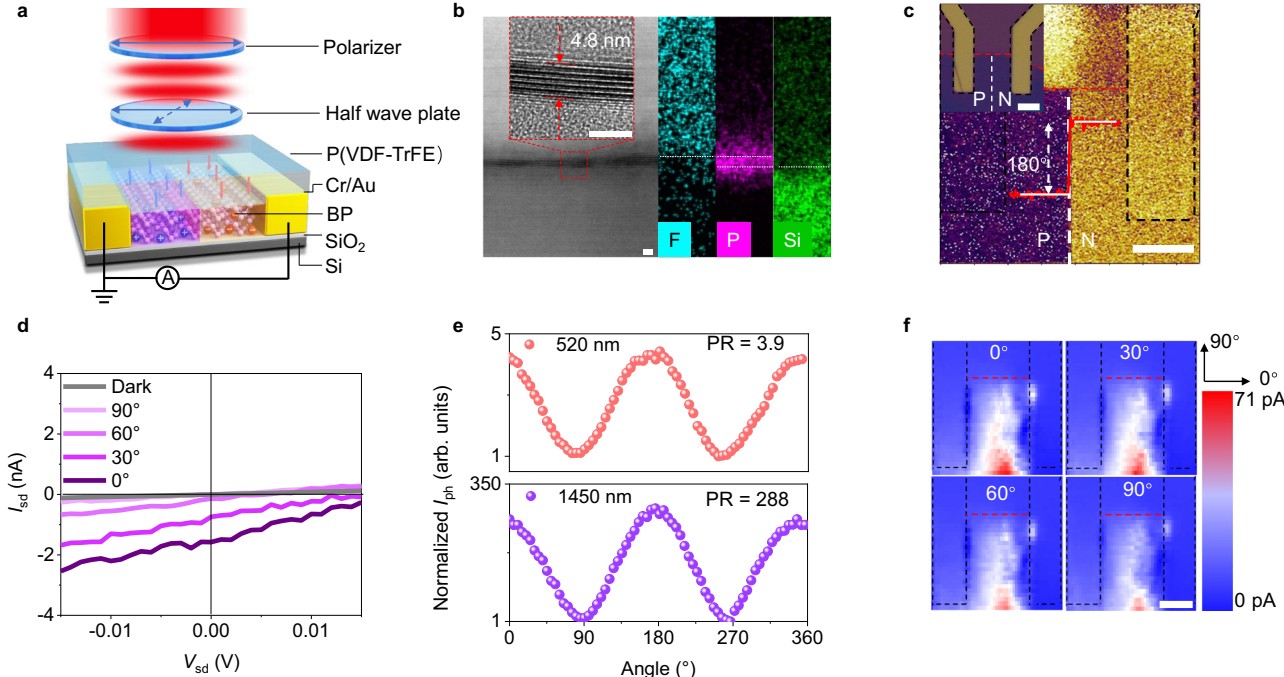

**Fig. 3 BP in-plane PN homojunction photodetector with ultrahigh polarization sensitivity. a** Schematic of BP in-plane PN homojunction defined by ferroelectric domains. **b** High-resolution transmission electron microscopy (HR-TEM) image and energy-dispersive X-ray spectroscopy (EDS) elemental mapping of F, P and Si. The inset is the partial enlarged drawing HR-TEM image and the thickness of BP is 4.8 nm. Scale bar is 5 μm. **c** The PFM phase image of the device. The phase difference between p-doped (P) region and n-doped (N) region is 180°. The inset shows corresponding optical image. The black dashed lines indicate the electrodes, the red dashed lines indicate the edge of BP material and the white dashed lines indicate the boundary of P and N region. Scale bar is 5 μm. **d** $I_{sd}$–$V_{sd}$ curves in the dark and under light of different incident polarization angle ($\lambda = 1450$ nm, $P_{eff} = 10$ μW). **e** Normalized photocurrent of the BP in-plane PN homojunction as a function of the polarization angle. The unity corresponds to photocurrent when angle is 90° or 270° at 520 nm (10 μW) and 1450 nm (10 μW). Polarizatio ratio (PR) is defined as the ratio of maximum and minimum polarization-dependent photoresponse. **f** the spatial photocurrent map is obtained by scanning a 520 nm light beam with power 1 μW and spot size 1 μm. Scale bar is 5 μm.

fluorine (F), phosphorus (P), and silicon (Si) correspond to P(VDF-TrFE)/BP/SiO$_2$ layers in the HR-TEM image. The ferro-electric film is up-polarized and down-polarized via PFM technology, and then BP is p-doped and n-doped by these ferroelectric domains. Figure 3c demonstrates the phase image of the device and the corresponding optical image is shown in the inset. The phase difference along the red dashed line between $P_{up}$ state (p-doped region) and $P_{down}$ state (n-doped region) reveals that the ferroelectric film is patterned into two opposite domains with 180° domain wall. More detailed information about P(VDF-TrFE) polarized by PFM can be found in Supplementary Figs. 4-5. By measuring the transfer curves, the Debye length in the up-polarized state and down-polarized state are 88 nm and 55 nm, respectively. Thus, the ferroelectric field can fully modulate this BP of dozens of nanometers. The detailed calculation process is provided in Supplementary Fig. 6.

The $I_{sd}$–$V_{sd}$ characteristics of the BP in-plane PN homojunction were measured under 300 K and 1450 nm laser illumination, as shown in Fig. 3d. The photocurrent increases with the polarization angle changes from 90° to 0°, which conforms to the linear dichroism of BP[33]. The maximum short-circuit current ($I_{sc}$) and open-circuit voltage ($V_{oc}$) obtained at 0° are 1.58 nA and 0.016 V, respectively. The $I_{sc}$ and $V_{oc}$ measured in this work are at the same level with BP homojunction defined by the local electrostatic gating[34]. Figure 3e shows the dependence of the device photocurrent on the polarization angle of the incident light, where the wavelength of the incident light is 520 nm and 1450 nm. The PR is 3.9 at 520 nm, which is a slight increase compared to that of pristine BP. Excitingly, the device shows ultrahigh polarization sensitivity at 1450 nm with a PR as high as

288. The ultrahigh polarization sensitivity can be attributed to the superimposed effect of PTE and PV effect. For PTE effect, the carrier mobility ratio is enhanced because of the enhanced anisotropy of the band structure. According to the polarization Raman spectra, the electron-phonon interaction in the AC direction is enhanced by the remanent polarization field, indicating the reduced thermal conductivity in the AC direction. Thus, the PTE photocurrent ratio increases. For PV effect, the polarization ratio can be increased to one or two orders of magnitude. The built-in electric field can separate photo-generated electron-hole pairs and promote the PTE current to move directionally. The polarization photoresponse character-istics under other wavelengths are shown in Supplementary Fig. 7. The PR of the device increases with a longer wavelength, Because the wavelength is closer to the bandgap, the dichroism is more obvious[9]. The polarization photocurrent mappings with polarization angles of 0°, 30°, 60°, and 90° are shown in Fig. 3f. It is obvious that the photoelectric conversion mainly occurs at the ferroelectric domain wall and diffuses to the electrodes on both sides. This broad photocurrent mapping implies that there is another mechanism (PTE effect) to generate electron-hole pairs besides PV effect[22]. From the photocurrent mappings, the photocurrent can be regarded as a directional and global current like other photothermal devices and the built-in electric field determines the flow direction of photo-generated carriers.

## BP FET and BP homojunction defined by electrostatic gating.

To verify the enhancement effect of the external electric field on the polarization photoresponse of BP, we fabricated a BP FET

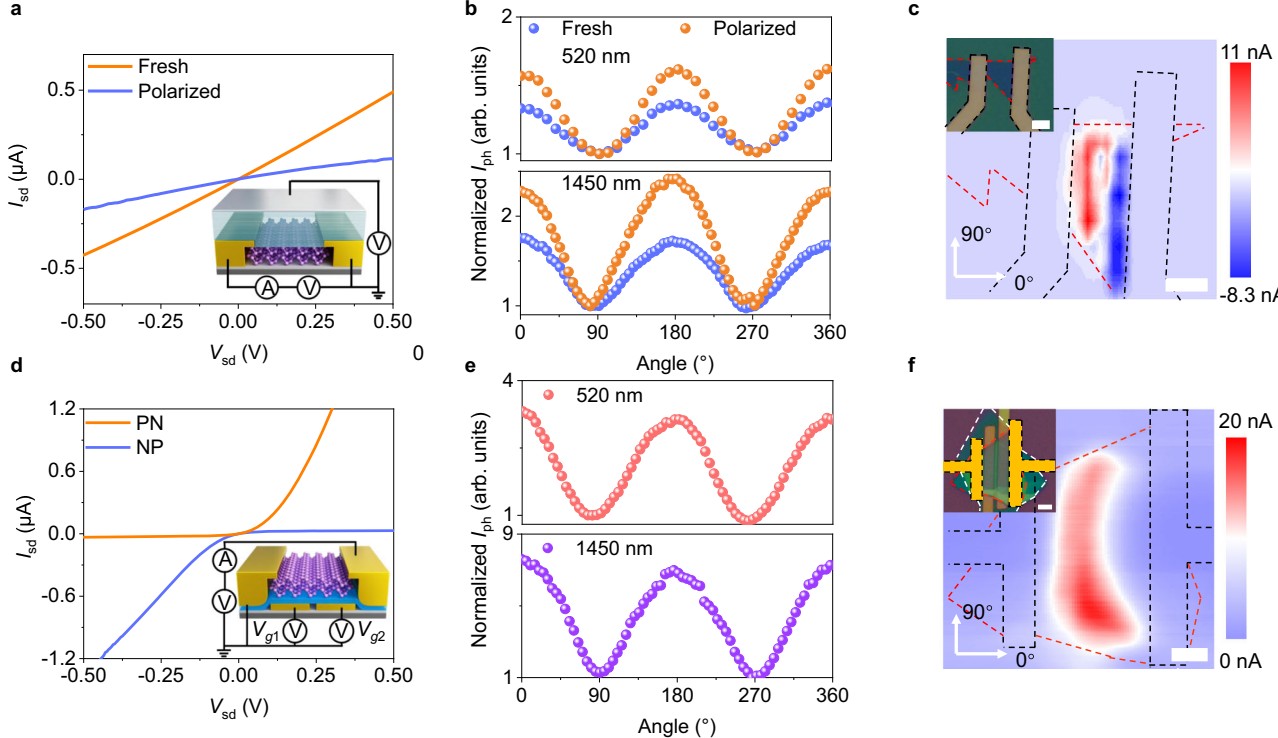

**Fig. 4 Electrical property and polarization photoresponse of BP FET and BP in-plane PN junctions defined by split bottom gates. a** Output characteristics of the BP FET. The inset shows the structure of device. 10 nm-Al was deposited as the gate electrode. **b** Normalized photocurrent of the BP FET as a function of the polarization angle at 520 nm (10 μW) and 1450 nm (10 μW). The unity corresponds to photocurrent when angle is 0° or 270°, $V_{sd} = 0.2$ V. **c** The spatial photocurrent map is obtained by scanning a 520 nm light beam with power of 10 μW, $V_{sd} = 0$ V. The inset is the optical image of device. The black dashed lines indicate the electrodes and the red dashed lines indicate the shape of BP material. Scale bar is 5 μm. **d** Output characteristics of BP PN junction and inset shows the BP is electrostatic doped into a PN junction (PN, $V_{g1} = -3$ V, $V_{g2} = 3$ V) and NP junction (NP, $V_{g1} = 3$ V, $V_{g2} = -3$ V). Few-layer hexagonal boron (h-BN) is hired as bottom gate dielectric. **e** Normalized photocurrent of the BP PN junction as a function of the polarization angle at 520 nm (10 μW) and 1450 nm (10 μW), $V_{sd} = 0$ V. The unity corresponds to photocurrent when angle is 90° or 270°. **f** Spatial photocurrent map obtained by scanning a 520 nm light beam with power of 20 μW and spot size of 1 μm, $V_{sd} = 0$ V. The inset is the optical image of device. The black dashed lines indicate the electrodes, the red dashed lines indicate the BP, the white dashed lines indicate the h-BN and the yellow dashed line reprensents practical scanning area. Scale bar is 5 μm.

with P(VDF-TrFE) as the gate dielectric (device structure is shown in the inset of Fig. 4a). Figure 4a shows the output characteristics of the device when the P(VDF-TrFE) is at the fresh and the polarization states. The symmetrical $I_{sd}$–$V_{sd}$ curves indicate that good Ohmic contact is realized. When P(VDF-TrFE) is polarized by the external gate voltage, the remanent polarization field restrained the carrier concentration in BP, and the drain current decreased at the same bias. Figure 4b demonstrates the polarization photoresponse characteristics of the device at the fresh and polarization states, where the incident light wavelength is 520 nm and 1450 nm ($V_{sd} = 0.2$ V), respectively. More detailed data is shown in Supplementary Fig. 8. When the device is in the fresh state, the PR is 1.36 at 520 nm and 1.72 at 1450 nm, while in the doped state, PR increases to 1.62 at 520 nm and 2.42 at 1450 nm. This enhancement polarization sensitivity is consistent with our previous deduction (the PRs of pristine BP at different wavelengths are shown in Supplementary Fig. 9). However, the large dark current in this device hinders its sensitivity to light. For the high carrier concentration of BP, even a small bias will lead to a large dark current. Although the device can work at zero bias with low dark current based on the Schottky junction, the opposite current flow is an obstacle to collecting photo-generated carriers. We performed the spatial photocurrent mapping of the device under zero bias, as shown in Fig. 4c, there are two obvious junctions between the BP and the electrodes. However, the photocurrent (PTE current) still transports at the opposite

direction, as analyzed in Fig. 1b, hindering the collection of photocurrent.

To explore the positive effect of PV effect in improving the polarization sensitivity of the device, we also fabricated a BP in-plane homojunction defined by local electrostatic gating. According to a previous report, a vertical BP homojunction enables higher sensitivity in polarized light detection[12]. Under illumination, the photo-generated electrons and holes in the PN junction are separated by the built-in electric field, thereby enhancing the polarization sensitivity of the photodetector. The few-layer BP used here is ambipolar and retains a slight p-doping property (as shown in Supplementary Fig. 6b), indicating that both hole and electron doping can be achieved by electrostatic doping[34]. Here, a BP in-plane homojunction was fabricated, and the gate dielectric is h-BN (the device structure is shown in the inset of Fig. 4d). In-plane PN junction ($V_{g1} = -3$ V, $V_{g2} = 3$ V) or NP junction ($V_{g1} = 3$ V, $V_{g2} = -3$ V) can be realized by applying opposite voltages to the two gates. As shown in Fig. 4d, this BP in-plane PN junction achieves a rectification of over 100 under a suitable gate voltage configuration (more output and transfer curves of this device can be found in Supplementary Fig. 10). When tuned into PN junction, this BP PN junction exhibits an obviously enhanced polarization sensitivity at zero bias. As shown in Fig. 4e, the PR is 3.44 at 520 nm incident light and 6.2 at 1450 nm incident light (more detailed data are shown in Supplementary Fig. 11). Figure 4f is a spatial photocurrent

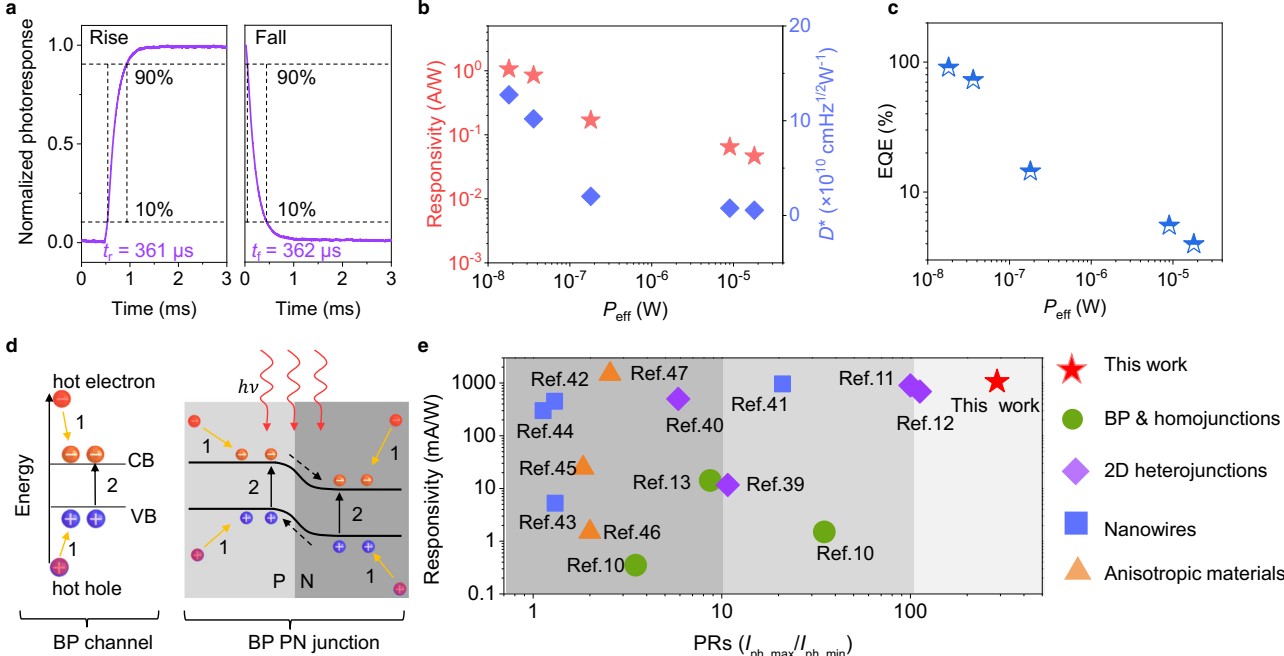

**Fig. 5 Optoelectronic measurement at a wavelength of 1450 nm. a** 90–10% rise ($t_r$) and decay ($t_f$) time of photocurrent are measured as 361 and 362 µs, respectively. **b**, **c** Calculated responsivity ($R$), detectivity ($D^*$), and external quantum efficiency (EQE) with effective incident optical powers ($P_{eff}$). **d** The carrier multiplication process in BP channel and BP PN junction. The orange and blue spheres represent electron and hole, respectively. The orange and black arrows represent step 1 and stpe 2. The black dashed arrows represent the flow direction of carriers. **e** Responsivity and PRs of BP in-plane PN homojunction defined by ferroelectric domains outperform most of previously polarization photodetectors, including BP, BP homojunctions defined by other methods, 2D heterojunctions, nanowires, and other anisotropic materials.

mapping at 520 nm incident light, and the inset is the optical image of the device. By comparing the optical image and photocurrent mapping image, it is unambiguously seen that the photon-generated carriers mainly appear in the PN junction area defined by the split bottom gates. This slit-shaped current mapping and non-linear output curve indicate it is a typical PN junction device, whose photoresponse is dominated by PV effect[22,35]. We do not rule out the PTE effect in this device for the photocurrent area is larger than the slit between a pair of split gates. However, it is the PV effect that dominates the photoresponse in this BP PN junction rather than the PTE effect (notably different from broad photocurrent mapping in Fig. 3f). It is probable that the strong PV effect introduces a large built-in electric field, forcing the photoelectric conversion to concentrate in the PN junction area and decreasing the photoresponse of the BP non-junction area.

Through the above design and mechanism analysis, the BP in-plane PN junction defined by ferroelectric domains exhibits excellent polarized light detection performance. Moreover, other figures of merits of the device have also been characterized in detail. The dynamic response under 1450 nm incident light, measured by switching on/off the mechanical chopper, is shown in Fig. 5a. As a photovoltaic device, this BP in-plane PN homojunction has a fast photoresponse speed with a rising time ($t_r$) of 361 µs and decay time ($t_f$) of 362 µs, which are shorter than that of BP photoconductive devices (usually several milliseconds or longer, as shown in Supplementary Fig. 9b). Compared with BP PN homojunctions fabricated via other approaches (Al-doping and chemical-doping), the device in this work also shows superior photoresponse speed[36,37]. The responsivity $R = I_{ph}/P_{eff}$ and detectivity $D^* = RA^{1/2}(2eI_{dark})^{-1/2}$ under various incident optical powers are calculated and shown in Fig. 5b. The device has a responsivity of up to 1.06 A/W and detectivity of up to $1.27 \times 10^{11}$ cm Hz$^{1/2}$W$^{-1}$, which is larger than most BP-based

photodetectors. To better evaluate the performance of the device, we also measured spectral-noise density (Supplementary Fig 12). The $D^*$ is calculated as $1 \times 10^{11}$ cm Hz$^{1/2}$W$^{-1}$, which is in good agreement with the calculation result based on dark current. The external quantum efficiency (EQE) of the device is given by the expression EQE = $(I_{ph}/P_{eff})(hc/e\lambda)$ and shows a maximum of 90.8% under weak light signal conditions (Fig. 5c). Such a high efficiency may be attributed to the carrier multiplication by hot carriers in BP (Fig. 5d), which is prevailing in low-dimensional materials for their strong Coulombic interactions[38]. When under the illumination of photoexcitation at energy above the energy gap, photo-generated carriers appear at the elevated positions in the conduction band and valence band, resulting in intravalley scattering of the carriers by delivering the excess energy to phonons (step 1). The photo-generated carriers scatter and stimulate the additional electron across the bandgap to the conduction band, resulting in carrier multiplication (step 2). Carrier multiplication has been proposed to enhance the efficiency of photodetectors. Therefore, this device shows a large EQE. To further demonstrate the performance advantages of BP in-plane PN homojunction defined by ferroelectric domains, we compared it with BP[10,13], BP homojunction made by other methods[10], 2D heterojunctions[11,12,39,40], nanowires[41–44], and other materials with dichroism[45–47]. Our device has obvious advantages in both PRs and responsivity, are shown in Fig. 5e. The responsivity of the device in this work is at the advanced level. More importantly, our device processes ultrahigh polarization sensitivity than those devices reported previously (detailed information is shown in Supplementary Table1).

## Discussion
In summary, we have designed and fabricated a BP in-plane PN homojunction defined by ferroelectric domains with ultrahigh

polarization sensitivity at room temperature. By performing the polarized Raman spectroscopy measurements, we have analyzed and obtained the enhancement effect of the ferroelectric field on the electron-phonon interactions along the AC direction. The PTE current in the AC direction increases because of reduced thermal conductivity and enhanced carrier mobility, thereby increasing the polarization detection ratio. By introducing the PN junction with the upward and downward ferroelectric field, the photo-generated carriers are separated more efficiently with the action of the built-in electric field. Moreover, the PV effect also has a positive effect on improving the polarization detection ratio of the device. As both the PV effect and the enhanced PTE effect contribute to the photocurrent anisotropic transport process, this BP in-plane PN junction-based photodetector shows an ultrahigh polarization ratio of 288 at 1450 nm incident light and at room temperature and atmosphere. The responsivity of 1.06 A/W and the detectivity of $1.27 \times 10^{11}$ cm Hz$^{1/2}$W$^{-1}$ are at an advanced level among similar devices. This design principle might also apply to other anisotropic materials, providing a train of thought in designing high-performance polarization-resolved photodetectors. To abtain full polarization information, it is possible to envision an imaging system based on BP flakes with four orientation (0°, 45°, 90°, and 135°).

## Methods

**Device fabrication.** For the BP in-plane homojunction constructed by ferroelectric field and BP FET, few-layer BP samples were prepared by mechanical exfoliation method and transferred onto 285 nm-thick SiO$_2$/Si substrates. The source and drain electrodes (chromium/gold (Cr/Au), 15/35 nm) were deposited on the BP through standard e-beam lithography and metal deposition techniques. P(VDF-TrFE) film with 100 nm thickness was prepared by spin-coating method. For BP in-plane PN homojunction constructed by ferroelectric field, P(VDF-TrFE) were polarized via PFM technology. For BP FET, P(VDF-TrFE) was polarized by the top gate and the 10 nm-thick aluminum film was deposited by thermal evaporation and patterned by photolithography as the top gate transparent electrode. For BP PN junction defined by local electrostatic gating, a pair of split bottom gates (Cr/Au, 15/25 nm) was fabricated on a 285 nm-thick SiO$_2$/Si substrate by standard e-beam lithography and metal deposition techniques. 20 nm-thick h-BN was transferred on the gates. Then, few-layer BP was transferred on h-BN, and the source and drain electrodes (Cr/Au, 15/35 nm) were deposited. Then, poly(methyl methacrylate) (PMMA) film was coated on this device to protect it from oxygen and water.

**Materials characterization.** A Lab Ram HR800 from HORIBA spectrometer and an Olympus ×100 objective lens were used for Raman measurements. All the polarized Raman measurements were illuminated with a 514.5 nm wavelength laser of 1.25 mW power under ×100 objective. A linear polarizer (Thorlabs) was used for polarized Raman measurements. The TEM and EDS measurements were performed by JEOL JEM2100F TEM with EX-24063JGT EDS. AFM and PFM measurements were performed using PFM mode by Cypher S from Asylum Research.

**Electrical and optoelectronic characterization.** In this study, all electronic and optoelectronic measurements were performed at room temperature and under ambient conditions. Electronic measurements were conducted using a commercial Keysight B1500A. All the optoelectronic measurements were taken by a MStarter 200 optoelectronic measurement system from Maita Optoelectronic Technology Co., LTD.

## Data availability

All technical details for producing the figures are enclosed in the Supplementary Information. All other relevant data are available from the corresponding author upon request. Source data are provided with this paper.

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

## Acknowledgements

This work is supported by National Key Research and Development program of China (grant no. 2021YFA1200700 (J.W.)), National Natural Science Foundation of China (grant nos. 62025405 (J.W.), 61835012 (J.W.), 61905267 (X.W.), 62105100 (Y.C.), 62075228 (X.M.), 61974153 (T.L.), and 62011530043 (H.S.)), Key Research Program of Frontier Sciences, CAS (grant no. ZDBS-LY-JSC045 (X.W.)), Strategic Priority Research Program of Chinese Academy of Sciences (grant no. XDB44000000 (J.W.)), Hundred Talents Program of the Chinese Academy of Sciences (X.W.), Science Technology Commission of Shanghai Municipality (grant no. 2151103500 (J.W.)), and Shanghai Sailing Program (grant no. 21YF1454800 (Y.Z.)).

## Author contributions

S.W. and Y.C. contributed equally to this work. J.W., X.W., and J.C. conceived and supervised the research. S.W. and X.W. fabricated the devices. Q.Z., H.C., and X.-J.W. performed the Raman measurements. S.W., Y.C., X.H., H.J., and X.T. performed the electrical and optoelectronic measurements. S.W. and Y.Z. performed the PFM measurements. S.H. and H.Y. performed the infrared spectroscopy measurements. T.L., H.S., W.H., X.M., J.C., and J.W. advised on the experiments and data anylasis. S.W., Y.C., and X.W. co-wrote the paper. All authors discussed the results and revised the manuscript.

## Competing interests

The authors declare no competing interests.
