## [Peer Review File · Nature Communications]

Ultra-sensitive polarization-resolved black phosphorus
homojunction photodetector defined by ferroelectric domainsREVIEWER COMMENTS

Reviewer #1 (Remarks to the Author):

In this manuscript, Wu et al report an interesting result on black phosphorus (BP) homojunction constructed by ferroelectric domains. The PFM conducting probe polarizes the ferroelectric polymer (PVDF-TrFE) and provides remnant polarization electric fields in different directions (Pup and Pdown), which in turn doped BP into p-type and n-type. As a result, the device exhibits a high polarization ratio of 288. The authors additionally fabricated BP FETs and BP pn junctions defined by local electrostatic gating, which were used to resolve the detection mechanism of the devices under the influence of photovoltaic (PV) and photothermoelectric (PTE) effects. The reported data in this work are reliable and support the main conclusions. It may also be valuable for designing new polarization photodetectors. Based on these results, I would like to recommend it for publication in Nature Communications after addressing the following issues.

1. In Fig. 2, the data shows the Raman spectra of BP, but it is unclear why the Raman intensity is enhanced by the ferroelectric field. The authors should explain in more detail.
2. The authors fabricated two kinds of BP pn homojunctions in this work, one doped by ferroelectric field and the other doped through a pair of split gates. What is the influence of the two methods on device performance, and why is the rectification ratio of the latter device much larger?
3. The polarization ratio of 288 at 1450 nm is noticeable, is there any other device showing such a high polarization ratio?
4. In the device structure, the authors choose a 4.8 nm-thick BP and a 100 nm-thick P(VDF-TrFE) to fabricate the device. How does the thickness of BP and P(VDF-TrFE) affect the device performance, especially polarization sensitivity?
5. The origin of photoresponse in BP is not only the PTE effect but also the photo-bolometric effect. Does the photo-bolometric effect also contribute to the photoresponse in BP? If so, an explanation is necessary.
6. Why the photocurrent mapping in Fig. 3f show that the largest current value appears at the bottom of the source and drain electrodes?
7. Why choose the BP to build this pn junction and what are its advantages? Could this approach also enable polarization detection for non-anisotropic 2D materials?

Reviewer #2 (Remarks to the Author):

Wu et al. report the use of ferroelectric materials to enhance the polarization sensitivity of black phosphorus photodetectors in the infrared range. The polarization ratio reaches 288, which is impressive in 2D materials-based polarization detectors. The responsivity and detectivity are also satisfying, spelling out the potential for practical applications. I have no concerns about the novelty, broad interest, and performance. I think it can be published in a high-impact journal like Nature Communications if the authors can address my concerns as below:

1. The mechanism of the ultrahigh polarization ratio ~ 288 is still not clear to me. The authors mentioned PTE effect, PV effect, and phonon drag effect, but I cannot find a clear description of how the $PR=288$ is achieved. A statement such as "The ultrahigh polarization sensitivity is attributed to the enhanced electron-phonon interaction in the AC direction by the remanent polarization field" is really vague and not well supported. To my understanding, the large PR should come from the dichroic absorption of BP when external bias (like FE field) is applied. If so, why the similar measurement of PR in Figure 4 and Supp Figure S8, S9 are much smaller, although their dichroic absorption should be the same? I guess the actual mechanism could be the interplay of PTE and PV effects, which may cancel each other in a certain way, or others?
2. The presentation can be improved, since many discussion in this work is really qualitative. For example, it seems that Figure 1 does not contain any simulated or measured results. The authors may consider the provide first-principle calculation results for the band diagram.
3. How did the authors estimate or measure the noise? Note that noise is dependent on the

working frequency and power density. Noise spectra and light noise would better help to evaluate the performance and justify their claim of "ultrasensitive".

4. How is the authors' approach based on anisotropic materials compared with other approaches such as artificial nanostructures (Nat Commun 11, 6404, 2020 from Prof Qiu group)? In this NC paper, the polarization can be detected even without power calibration due to their intriguing vectorial photoresponse. Besides, ref 8 also reports very high PR and even negative PR. To my understanding, the nanostructures-based approach should be more useful in practice, since their orientation can be controlled lithographically. Instead, I feel it is challenging for BP to orient along all 0, 45, and 90 degrees within the same flake to acquire complete polarization information. In a broader scope, how is this approach compared with metasurface ones (ref 1 in the manuscript from Prof Capasso group)? The authors may consider including a general discussion rather than just presenting another work in the BP community.

5. Have the authors checked the photoresponse of the ferroelectric materials used in their device? Note that FE materials usually show bulk photoresponse with very strong polarization dependence. The PR can even be negative. Potential BPVE should be excluded.

6. The authors should double-check the data from other papers. For example, the PR of ref 15 (Nat. Photonics 12, 601-607, 2018) is more than 100, but labeled as 30 in Figure 5e.

7. Abstract Line 23 "the performance of these devices is restricted by intrinsic property of materials, including crystal structure, electrical conductivity, and dichroism". To my understanding, the polarization ratio should have no relation with the anisotropic electrical conductivity?? Although the anisotropic mobility of BP also originates from the BP crystal symmetry, once the device is fabricated, the charges flow only in the direction along the channel. Besides, dichroism is a result of anisotropic "crystal structure", no need to repeat.

8. Page 12 Line 309, "However, it is the PV effect that dominates the photoresponse in this BP PN junction rather than the PTE effect (notably different from photocurrent mappings in Fig. 3f)." I do not understand why PV is dominant over PTE. Citation needed? Besides, the authors should explain how the results are different from Fig 3f, and how to interpret the difference.

Reviewer #3 (Remarks to the Author):

The submitted manuscript "Ultra-sensitive polarization-resolved black phosphorus in-plane homojunction constructed by ferroelectric domains", by Wu et al propose a BP homojunction photodetector with ultrahigh polarization sensitivity. On the basis of analysis and experimental results, the authors have enhanced the polarization ratio of BP photodetectors a lot and very carefully explain their work mechanism (PV and PTE effect). According to this manuscript, the ferroelectric field of P(VDF-TrFE) is a useful mean of regulating BP's photothermoelectric properties as well as for designing device structure. The result is reliable and interesting because it proves that the anisotropy of BP and even other materials with anisotropic crystal structure has tunable degrees of freedom. Therefore, it is worthy publication in Nature Communications. There are some minor comments that need be addressed before publication.

1. Fig 2 shows that ferroelectric field enhances Raman intensity of BP. Is this phenomenon specific in P(VDF-TrFE)?

2. The authors also studied the device tuned by electrostatic field in Fig 4a, does the Raman intensity of this device change?

3. Fig 4f shows that thick BP has better light response. Why didn't the author choose thicker material?

4. The photocurrent based on PTE effect is usually attributed to the local heats resulting from the nonuniform illumination. The illumination on BP device seems to be uniform. Can the author explain this for more details?

Response to Reviewers' Comments

Reviewer 1:

Comments: In this manuscript, Wu et al report an interesting result on black phosphorus (BP) homojunction constructed by ferroelectric domains. The PFM conducting probe polarizes the ferroelectric polymer (PVDF-TrFE) and provides remnant polarization electric fields in different directions (P_{up} and P_{down}), which in turn doped BP into p-type and n-type. As a result, the device exhibits a high polarization ratio of 288. The authors additionally fabricated BP FETs and BP pn junctions defined by local electrostatic gating, which were used to resolve the detection mechanism of the devices under the influence of photovoltaic (PV) and photothermoelectric (PTE) effects. The reported data in this work are reliable and support the main conclusions. It may also be valuable for designing new polarization photodetectors. Based on these results, I would like to recommend it for publication in Nature Communications after addressing the following issues.

Answer: We would like to thank the reviewer for his/her carefully reading the manuscript and providing positive comments. We thank the reviewer for his/her sincere recommendations, such as “It may also be valuable for designing new polarization photodetectors”. We believe the comments proposed by the reviewer are very helpful to improve our manuscript. The issues have been addressed and answered one by one as follows:

1. In Fig. 2, the data shows the Raman spectra of BP, but it is unclear why the Raman intensity is enhanced by the ferroelectric field. The authors should explain in more detail.

Answer: Thank you for your valuable comment and suggestion. Raman spectroscopy is a fast and nondestructive characterization tool that provides structural and chemical information of materials. In recent years, Raman spectroscopy has been widely applied in characterizing 2D layered materials. However, the weak intensity of Raman scattering light limits the application of Raman scattering spectroscopy at initial

stage. Surface-enhanced Raman scattering (SERS) was first observed by Fleischmann et al on a roughened silver electrode (Chemical physics letters, 1974, 26(2), 163). The origin of SERS enhancement is related to the electromagnetic or chemical effect. In our work, the enhanced Raman intensity is obviously related to electromagnetic effect, which plays a dominant role in SERS. The electromagnetic enhancement comes from the local electric field enhancement and re-radiation enhancement. Here, the polarized ferroelectric film P(VDF-TrFE) provides a strong local ferroelectric field on BP surface. As a result, the Raman intensity of BP is enhanced by this ferroelectric field. A noteworthy property of BP crystals is the dependence of the Raman mode intensities on the polarization of incident and scattered light. Therefore, the Raman intensity is enhanced anisotropically.

2. The authors fabricated two kinds of BP pn homojunctions in this work, one doped by ferroelectric field and the other doped through a pair of split gates. What is the influence of the two methods on device performance, and why is the rectification ratio of the latter device much larger?

Answer: Thanks for your careful reading and kind comments. We fabricated two BP homojunction devices by polarized P(VDF-TrFE) and a pair of split gates. The differences between the two kinds of BP photodetectors are summarized as follow:

Figure R1-1. Differences between two devices.

As we mentioned in the manuscript, the photoresponse mechanism of device 1 is the coupling effect of PV and PTE, while the dominant photoresponse mechanism of device 2 is the PV effect. With the PV effect, the dark current can be reduced, the response speed can be improved and polarization ratio can be increased. With the PTE

effect, the spectral response range can be broadened, and the polarization ratio also can be increased. In this work, we mainly focus on the polarization sensitivity of device. Therefore, for device 1, the combination of PV and PTE effects improves the polarization ratio efficiently. The device 2 possesses a large rectification ratio of 279, much larger than that of device 1 (1.96). This is due to the depolarization field of P(VDF-TrFE). The depolarization effect caused by the interface characteristics will reduce the rectification ratio of the black phosphorus homojunction to a certain extent. However, once the polarization field is stabilized, the device performance will also remain stable.

3. The polarization ratio of 288 at 1450 nm is noticeable, is there any other device showing such a high polarization ratio?

Answer: Thank you for your helpful comment. We performed some additional experiments, and the results are shown in Figure R1-2. After polarizing P(VDF-TrFE) by the PFM conductive probe, we measured the photocurrent versus polarization angle for two devices under 1450 nm incident light. For device 1, the highest polarization ratio is 114. For device 2, the highest polarization ratio is 122. Based on these experimental results, the method proposed in our work is repeatable and reliable.

Figure R1-2. Normalized photocurrent of the BP in-plane PN homojunction as a function of the polarization angle of device 1 (a) and device 2 (b). The unity corresponds to photocurrent when angle is 90° or 270° at 1450 nm.

4. In the device structure, the authors choose a 4.8 nm-thick BP and a 100 nm-thick P(VDF-TrFE) to fabricate the device. How does the thickness of BP and P(VDF-TrFE) affect the device performance, especially polarization sensitivity?

Answer: Thanks for your kind comment. BP exhibits a thickness-tunable direct bandgap, from 1.5 eV for monolayer and 0.3 eV for bulk. Thus, the thickness of BP will decide its work waveband. When used in photodetectors, BP is supposed to be thick for better optical absorption. From an electrical standpoint, the thickness of BP should be thin to reduce noise and improve the carrier extraction efficiency. There are some reports show that a vertical electric field can effectively reduce the bandgap of few-layer BP (Nature communications, 2017, 8(1): 14474, Science advances, 2020, 6(7): eaay6134.). Therefore, we mainly choose thin BP in our work. To investigate the influence of BP's thickness on our device performance, we performed some supplementary experiments. As shown in Figure R1-3, we choose BP with different thicknesses to fabricated four devices. With thickness from thin to thick, the PR values change from 122 to 15. This can be attributed to that the intrinsic carrier concentration of BP film decreases with reducing thickness (ACS Photonics, 2017, 4(7): 1822). The higher carrier concentration of thick BP weakens the tunable effect of ferroelectric field, leading high dark current and small polarization ratio. Therefore, it is better to choose thin BP in this work.

Figure R1-3. Normalized photocurrent of the BP in-plane PN homojunction as a function of the polarization angle of BP photodetector with different thickness from (a) to (b).

As for P(VDF-TrFE), it has good transmittance in the visible to near-infrared band. Moreover, we studied that its remanent polarization is almost invariant with thickness (npj 2D Materials and Applications 1(1): 38). Here, we choose 100 nm thick P(VDF-TrFE) that can provide a strong enough electric field to tune few-layer BP. It is also considered that P(VDF-TrFE) with a certain thickness can protect the black phosphorus from the influence of the environment.

5. The origin of photoresponse in BP is not only the PTE effect but also the photo-bolometric effect. Does the photo-bolometric effect also contribute to the photoresponse in BP? If so, an explanation is necessary.

Answer: Thank you very much for your valuable comment. As you mentioned, the photocurrents in BP photodetectors are dominated by thermally driven thermoelectric and bolometric processes (Physical Review B, 2014, 90(8): 081408.). The photo-bolometric current will be generated across the whole device when applied a bias. As we described in the manuscript (Page 5, Line 101-104), the photothermoelectric current dominate the photoresponse of device when at zero or small bias. In view of our devices work at zero bias, the photo-bolometric effect can be ignored.

6. Why the photocurrent mapping in Fig. 3f show that the largest current value appears at the bottom of the source and drain electrodes?

Answer: Thank you for your careful reading and helpful comment. The inconsistent photocurrent results from the non-uniform thickness of BP in the channel. The BP at the bottom of the source and drain is thicker than other area. The thicker semiconductor has better optical absorption ability thus generates larger photocurrent. The non-uniform thickness in the exfoliated 2D materials is inevitable and sometimes it can't be identified from a microscope photograph.

7. Why choose the BP to build this pn junction and what are its advantages? Could this approach also enable polarization detection for non-anisotropic 2D materials?

Answer: We would like to thank the reviewer again for providing positive comment and valuable suggestion. BP offers an intrinsic crystal anisotropy for polarization-sensitive photodetection across a broad spectral range from visible to mid-infrared wavelength. Moreover, the strongly puckered structure renders its electronic state

highly tunable to external perturbations. The bipolar characteristics makes BP easily form a lateral PN junction.

We thank the reviewer for his/her kindly suggestion about the universality of the approach of the polarization-sensitive enhancement. Here, we performed supplementary experiments and applied this approach to MoTe₂ as shown in Fig. R1-4. It is worth noting that the MoTe₂ is a non-anisotropic semiconductor. However, the MoTe₂ PN homojunction defined by ferroelectric domains is also show polarization-resolved photocurrent. Therefore, this approach can be widely used to fabricate polarization-sensitive photodetectors.

Figure R1-4. (a) The PFM phase image of the MoTe₂ PN homojunction device. The phase difference between P region and N. The inset shows corresponding optical image. (b) Output characteristics of MoTe₂ FET with fresh state and polarized state. Normalized photocurrent of the MoTe₂ PN homojunction as a function of the polarization angle of device under the illumination of (c) 830 nm, and (d) 1064 nm.

Reviewer 2:

Comments: Wu et al. report the use of ferroelectric materials to enhance the polarization sensitivity of black phosphorus photodetectors in the infrared range. The polarization ratio reaches 288, which is impressive in 2D materials-based polarization detectors. The responsivity and detectivity are also satisfying, spelling out the potential for practical applications. I have no concerns about the novelty, broad interest, and performance. I think it can be published in a high-impact journal like Nature Communications if the authors can address my concerns as below:

Answer: We would like to thank the reviewer for the kind comments “The polarization ratio reaches 288, which is impressive in 2D materials-based polarization detectors.”. We would also like to thank the reviewer for recommending this work for publication in Nature Communications. Moreover, we especially appreciate the reviewer for his/her professional questions on the sensitivity and those suggestions for further improving the manuscript. We have addressed the concerns and replied to the comments one by one as follows.

1. The mechanism of the ultrahigh polarization ratio ~ 288 is still not clear to me. The authors mentioned PTE effect, PV effect, and phonon drag effect, but I cannot find a clear description of how the $PR=288$ is achieved. A statement such as “The ultrahigh polarization sensitivity is attributed to the enhanced electron-phonon interaction in the AC direction by the remanent polarization field” is really vague and not well supported. To my understanding, the large PR should come from the dichroic absorption of BP when external bias (like FE field) is applied. If so, why the similar measurement of PR in Figure 4 and Supp Figure S8, S9 are much smaller, although their dichroic absorption should be the same? I guess the actual mechanism could be the interplay of PTE and PV effects, which may cancel each other in a certain way, or others?

Answer: Thank for careful reading and kind comment. We are sorry to have confused you with the description of the polarization detection mechanism of the device. As you point out, the dichroic absorption of BP enables BP-based photodetectors to detect polarized light, the enhancement of the polarization ratio to 288 here is the result of the

interaction of PTE and PV effects. Here, we attempt to clearly explain the mechanism by which the above two effects enhance the polarization ratio of the device.

For PTE effect, the efficiency of the thermoelectric material (ZT) is calculated by $ZT = \frac{S^2\sigma T}{K}$, and is positively related to the Seebeck coefficient (S). There have been some reports on tuning the BP Seebeck coefficient by an external voltage. According to the calculation results of Li Yang et al, the ZT along the armchair direction can be greatly enhanced by the electric field, while the ZT along the zigzag direction is almost unchanged (Nano letters, 2014, 14(11): 6393). Therefore, the polarization ratio of the PTE current introduced by polarized light can be increased by the electric field. Although we were not able to directly measure the Seebeck coefficient of BP in this work due to the limitations of our experimental conditions, we can still infer that the Seebeck coefficient of BP is enhanced by the ferroelectric field through the interaction of electrons and phonons. Phonons play a primary role in semiconductors and insulators, and the phonon drag effect is supposed to contribute to increasing the Seebeck coefficient by phonon diffusion. By performing polarization Raman spectra measurements, we confirm that the electron-phonon interaction intensity is enhanced by the ferroelectric field in the armchair direction. Therefore, we conclude that when the device is illuminated by 0° polarized incident light, it has a larger PET current, which in turn increases the polarization ratio. As you noticed, the ferroelectric field does not change the dichroic absorption of BP. The PR enhancement of the device in Figure 4a is the result of the ZT being enhanced in the armchair direction.

Besides, for the PV effect, there have also been many reports on enhancing PR by fabricating PN junction-based devices, such as BP/MoS₂ (Nature Photonics, 2018, 12(10): 601), BP/WSe₂ (Nano Energy, 2017, 37: 53), and BP PN homojunction (Nature nanotechnology, 2015, 10(8): 707). In this work, we fabricated a black phosphorus in-plane PN junction based on two split gates. From Figure 4d in the manuscript, it can be seen that the PR of the device is enhanced by the photovoltaic effect. We consider that the photovoltaic effect of the PN junction directionally regulates the transport behavior of photogenerated carriers to a certain extent, and improves the extraction efficiency of

photogenerated carriers.

To sum up, we conclude that the actual enhancement mechanism of the polarization ratio of the device is the synergistic effect of PTE and PV effect. Correspondingly, we have added a clear description in the revised manuscript (Page 10, Line 247-256), hoping to be approved by the reviewer.

2. The presentation can be improved, since many discussion in this work is really qualitative. For example, it seems that Figure 1 does not contain any simulated or measured results. The authors may consider the provide first-principle calculation results for the band diagram.

Answer: Thank you very much for your valuable comments. We agree with you that first-principle calculation can provide quantitative band energy information. Not only that, there have been some theoretical and experimental results focusing on the modulation of the BP band structure, which is of great significance for the study of BP-based polarization detectors. Jimin Kim, et al (Science, 2015, 349(6249): 723) have demonstrated that under the modulation of a vertical electric field, the BP turns into a Dirac semimetal with anisotropic dispersion, which is linear in the armchair and the quadratic in zigzag directions. With further increasing in dopant density, E_g gradually reduces (Fig. R2-1) and then becomes zero, the angle-resolved photoemission spectroscopy (ARPES) spectra taken along the armchair direction (k_x) becomes linear (Fig. R2-1H), whereas that in the zigzag direction k_y remains nearly parabolic. Band structures are measured by ARPES, and they also provide DFT calculations. Moreover, Yanpeng Liu et al. (Nano Lett. 2017, 17, 1970–1977) observed a huge Stark effect in electrostatically gated few-layer BP using cryogenic scanning tunneling microscopy. As shown in Fig. R2-2, the gate voltage-dependent scanning tunneling spectroscopy (STS) acquired in the defect-free region of the 11-layer BP flakes. The dI/dV spectra associated with the gate modulation are in agreement with the Perdew-Burke-Ernzerhof parametrization (PBE) band structure calculations.

Given all of that, the band structure of BP under chemical doping and electrostatic gating has been intensively studied and reported. On the basis of these solid studies, it is reasonable for us to design the device with the conclusion that the external electric

field can tune the BP band structure. Furthermore, we focus on the anisotropic properties of BP, and the above theory and experiments also can support the plausibility of our delineated band structure.

Figure R2-1. Band structure and tunable E_g of few-layer BP. (Science, 2015, 349(6249): 723)

Figure R2-2. Gate-controlled Stark effect in an 11-layer BP flake device measured by scanning tunneling microscopy. (Nano Lett. 2017, 17, 1970–1977)

3. How did the authors estimate or measure the noise? Note that noise is dependent on the working frequency and power density. Noise spectra and light noise would better help to evaluate the performance and justify their claim of “ultrasensitive”.

Answer: Thank for your helpful suggestion. To better evaluate the performance of our devices, we added noise spectra and the result is shown in Fig. R2-3. The specific detectivity D^* is defined as $D^* = \frac{\sqrt{A\Delta f}}{NEP} = \frac{R\sqrt{A\Delta f}}{I_n}$, where A is the active area of device, I_n is the noise current, and Δf is the measurement bandwidth. The D^* is calculated as $1 \times 10^{11} \text{ cmHz}^{1/2}\text{W}^{-1}$, which is comparable to our previous calculation result ($1.27 \times 10^{11} \text{ cmHz}^{1/2}\text{W}^{-1}$) based on dark current. The spectral-noise density is added to the revised Supplementary Information (Supplementary Fig. 12). We also add some discussion to the revised manuscript (Page 13, Line 333-336).

Figure R2-3. Spectral-noise density of the device at 300 K and zero bias.

4. How is the authors’ approach based on anisotropic materials compared with other approaches such as artificial nanostructures (Nat Commun 11, 6404, 2020 from Prof Qiu group)? In this NC paper, the polarization can be detected even without power calibration due to their intriguing vectorial photoresponse. Besides, ref 8 also reports very high PR and even negative PR. To my understanding, the nanostructures-based approach should be more useful in practice, since their orientation can be controlled lithographically. Instead, I feel it is challenging for BP to orient along all 0, 45, and 90 degrees within the same flake to acquire complete polarization information. In a broader scope, how is this approach compared with metasurface ones (ref 1 in the manuscript

from Prof Capasso group)? The authors may consider including a general discussion rather than just presenting another work in the BP community.

Answer: Thanks for your thoughtful and meaningful suggestion. Metallic nanostructures can spatially confine light at the nanoscale to achieve its plasmonic resonances, leading to strong near-field intensity enhancement. Many optical nanoantennas, such as plasmonics, dielectric nanoantennas and optical cavities, are fabricated on 2D materials to enhance light absorption and emission. In addition to enhancing the light absorption of the material, metal nanoantennas can also regulate the carrier transport by controlling its shape. Qiu et al proposed a polarization photodetector with Bulk photovoltaic effect (BPVE) by fabricating non-centrosymmetric metallic nanoantennas on the surface of graphene (Nature communications, 2020, 11(1): 6368). Subsequently, they demonstrated a nanoantenna-mediated graphene polarization photodetector, whose PR value can be changed from positive to negative by tuning the orientation of the nanoantennas (Nature Photonics, 2021, 15(8): 614). Their excellent works not only obtained ultrabroad tunable graphene polarization photodetectors, but also provided new ideas for the promotion and development new polarization detection technologies.

It is no doubt that nanostructure-based approaches are stable and nonvolatile in practice devices, especially for large-area devices. However, the method proposed in our work also has the advantages of recommendation. First, P(VDF-TrFE) thin films can be easily fabricated onto 2D materials by spin coating and combined by van der Waals forces to form an excellent interface. Second, the nanostructure-based devices usually work at some specific wavelengths because it is determined by the parameters of the metal nanoantennas. And the operating wavelength in our work is determined by the material. Third, our method shows advantages in terms of non-destructiveness and editability, as it is easy to control the ferroelectric domains by external electric field.

Besides, as you mentioned, although it is difficult to acquire the complete polarization information of BP along 0° , 45° and 90° , we can identify the crystal orientation of BP before fabricating the device, such as by measuring angle-resolved polarization Raman spectroscopy and polarization-resolved infrared spectroscopy.

These measurements are also performed in our work and help to design our devices.

Capasso et al fabricated diffraction gratings by dielectric metasurfaces and demonstrate a full-stokes polarization camera (Science, 2019, 365(6448): eaax1839). They raised a matrix Fourier optics to realize the spatial variation of the polarization of light, providing a new strategy for polarization optics. In contrast to metasurface methods, the method in our work focuses on regulating the properties of the material instead of light modulation. However, it is referentially valuable for tuning 2D materials though patterning ferroelectric domains arrays.

Considering your wonderful suggestion, we have added a general discussion in the introduction of the manuscript to enhance the universal application and comprehensiveness (Page 4, Line 70-82, Reference 14 and 15).

5. Have the authors checked the photoresponse of the ferroelectric materials used in their device? Note that FE materials usually show bulk photoresponse with very strong polarization dependence. The PR can even be negative. Potential BPVE should be excluded.

Answer: Thank you very much for your valuable comment. We agree with you that ferroelectrics usually show bulk photoresponse with very strong polarization dependence (Nature nanotechnology, 2010, 5(2): 143; Nature communications, 2017, 8(1): 281). For materials without inversion symmetry, the bulk photovoltaic effect (BPVE) can also occur and generate electrical power without a junction (Science, 2021, 372(6537): 68). According to the reference (Nature, 2019, 570(7761): 349) as shown in Fig. R2-4, the photovoltaic effect and BPVE show different I–V characteristics in the dark and under illumination. In Fig. 3d, the output characteristics of our device show a typical photovoltaic response (as shown in Fig. R2-4(b)). In addition, the bulk photovoltaic photocurrent is along with the ferroelectric polarization direction, here perpendicular to the BP channel. Therefore, we think the BPVE can be excluded in our device.

Figure R2-4. (a) Comparison of three different light-to-current conversion mechanisms. (b) The output curves in the dark and under light of different incident polarization angles.

6. The authors should double-check the data from other papers. For example, the PR of ref 15 (Nat. Photonics 12, 601-607, 2018) is more than 100, but labeled as 30 in Figure 5e.

Answer: Thanks for careful reading and pointing out the mistake data in the manuscript. We made a correction in the manuscript (Page 4, Line 74), Figure 5e and supplementary Table 1.

7. Abstract Line 23 “the performance of these devices is restricted by intrinsic property of materials, including crystal structure, electrical conductivity, and dichroism”. To my understanding, the polarization ratio should have no relation with the anisotropic electrical conductivity?? Although the anisotropic mobility of BP also originates from the BP crystal symmetry, once the device is fabricated, the charges flow only in the direction along the channel. Besides, dichroism is a result of anisotropic “crystal structure”, no need to repeat.

Answer: We are very grateful for pointing out the defectives in our manuscript. We are sorry for the ambiguous expression in the abstract, and we have revised the sentence as follows: “the performance of these devices is restricted by intrinsic property of materials, including crystal structure and electronic states.” (Page 2, Line 22-24).

8. Page 12 Line 309, “However, it is the PV effect that dominates the photoresponse in this BP PN junction rather than the PTE effect (notably different from photocurrent mappings in Fig. 3f).” I do not understand why PV is dominant over PTE. Citation needed? Besides, the authors should explain how the results are different from Fig 3f, and how to interpret the difference.

Answer: Thank you for your careful reading and valuable suggestion. The photocurrent directions induced by the PTE and PV effects are the same, and the maximum photocurrent appears when the light is located at the junction interface. There are several basic criteria that can be used to identify photoresponse mechanisms. First, PV devices rely on the built-in electric field, which results in nonlinear output curves, while PTE devices possess linear output curves. Second, the photoelectric conversion in PV devices occurs around the junction interface and the photocurrent mapping is narrow, while the photocurrent mapping in PTE devices is quite broad (Advanced Materials, 2019, 31(50): 1902044). Considering the nonlinear output curves and narrow photocurrent mapping in Fig. 4d and f, we think that the PV effect dominates the photoresponse of device in Fig. 4d. The photocurrent mapping in Fig. 3f shows much broader than the junction region (Fig. 3c), and the output curve is linear. Therefore, the photoresponse mechanism of device in Fig. 3 is the synergistic effect of PTE and PV effect. Additional explanations have been added to the revised manuscript (Page 10, Line 262, Page 12, Line 311-312).

Reviewer 3:

Comments: The submitted manuscript “Ultra-sensitive polarization-resolved black phosphorus in-plane homojunction constructed by ferroelectric domains”, by Wu et al propose a BP homojunction photodetector with ultrahigh polarization sensitivity. On the basis of analysis and experimental results, the authors have enhanced the polarization ratio of BP photodetectors a lot and very carefully explain their work mechanism (PV and PTE effect). According to this manuscript, the ferroelectric field of P(VDF-TrFE) is a useful mean of regulating BP’s photothermoelectric properties as well as for designing device structure. The result is reliable and interesting because it proves that the anisotropy of BP and even other materials with anisotropic crystal structure has tunable degrees of freedom. Therefore, it is worthy publication in Nature Communications. There are some minor comments that need be addressed before publication.

Answer: We appreciate the reviewer’s positive comments on our work about the polarization detection performance enhancement and clear mechanisms clarification. In the following, we would like to provide detailed answers to each of your comments.

1. Fig 2 shows that ferroelectric field enhances Raman intensity of BP. Is this phenomenon specific in P(VDF-TrFE)?

Answer: Thanks for your comment. This phenomenon is not limited to the case of P(VDF-TrFE). Raman scattering can characterize the inelastic scattering of photons that can occur when they interact with matter. Inelastic scattering is an intrinsically very weak phenomenon whose Raman intensity can be enhanced by surface-enhanced Raman scattering (SERS). It is generally assumed that SERS arises from the electromagnetic and the chemical effects. The polarized P(VDF-TrFE) in this work provides a high ferroelectric field on the BP, resulting in enhanced Raman intensity. We also fabricated another BP device tuned by a vertical electric field and measured the Raman intensity of BP, as shown in Fig. R3-1. Apparently, the Raman intensity increases when a vertical electric field is applied on the BP.

Figure R3-1. Raman spectra of BP at fresh state and applied gate voltage.

2. The authors also studied the device tuned by electrostatic field in Fig 4a, does the Raman intensity of this device change?

Answer: Thank for your question. To determine whether the Raman intensity of the device in Fig. 4a changed, we fabricated new devices and performed Raman spectroscopy. As shown in Fig. R3-2(a), the intensity of three typical Raman peaks (A_g^2 , B_{2g} and A_g^1) increases when applied voltages on BP. Therefore, the electrostatic field can enhance the Raman intensity of BP. Fig. R3-2(b) extracts the peak intensity ratio of A_g^2 and A_g^1 at different gate voltages. When applied gate voltages on BP, the value of A_g^2/A_g^1 increased from 1.2 to 1.38. This phenomenon is in consistent with BP tuned by ferroelectric field, showing a tunable effect of electric field on the anisotropy of the BP.

Figure R3-2. Raman spectra of BP tuned by electrostatic field. (b) The ratio of Raman A_g^2 and A_g^1 peak intensity as a function of gate voltage.

3. Fig 4f shows that thick BP has better light response. Why didn't the author choose thicker material?

Answer: Thank for your comment. The BP device in Fig. 4f shows a larger photocurrent because of better optical absorption in thick materials. However, thick BP also possesses high carrier concentration, leading a large dark current. We also chose thick BP to fabricate photodetector and perform photoelectric performance measurements, as shown in Fig. R3-3. The output curves show high carrier concentration in thick BP and ferroelectric field can hardly tune the electricity of BP with such a thickness. The polarization ratio of this BP PN device is about 1.2, much smaller than that of thin BP device. Therefore, we choose thin BP to prepare device instead of thick one.

Figure R3-3. (a) Output curves of BP device. The inset are optical image and PFM phase image of device. (b) Normalized photocurrent of devices as a function of the polarization angle of BP photodetector.

4. The photocurrent based on PTE effect is usually attributed to the local heats resulting from the nonuniform illumination. The illumination on BP device seems to be uniform. Can the author explain this for more details?

Answer: Thank for your helpful comment. PTE detectors are based on the photothermal conversion and thermoelectric effect. Generally, the light illuminates on one side of PTE detectors, building up a temperature difference to drive charge carriers move from the hot end to the cold end and establishing an electric potential difference. This process is known as Seebeck effect and the Seebeck coefficient is strongly related to the electrical conductivity of the material. In this work, the BP is doped into n-type and p-type. The Seebeck coefficient gradient is formed by polarized P(VDF-TrFE), which can

drive carriers move directionally. Besides, the built-in electric field in BP PN junction also separates the photogenerated carriers. Therefore, even the device is in the uniform illumination, the photocurrent excited by PTE effect can be collected by electrodes.

REVIEWER COMMENTS

Reviewer #1 (Remarks to the Author):

All questions are well addressed, and recommend to be accepted for publication without change.

Reviewer #2 (Remarks to the Author):

Thanks for the authors' substantial efforts in answering my questions. While the manuscript has been greatly improved, I feel that some core concerns remain, and I cannot fully recommend the publication in its current form. Please see my comments below:

1. The mechanism is still not clear to me.

a. The authors have used the enhancement of ZT to explain the observed anisotropic PTE, but this is very strange, if not irrelevant. Note that the enhancement of ZT values usually rely on the suppression of the thermal conductivity of lattice while maintaining that of electrons, and this is why engineering phonons is a common approach to increase ZT. Look at the definition of ZT, then we understand that the increase of ZT may not necessarily be due to the increase of Seebeck coefficient (S), and in fact usually not the case. Therefore, the authors should carefully check their analysis and avoid being illogical.

b. Even we assume that the Seebeck coefficient can be indeed enhanced along the armchair direction, I still don't understand how the PR can be enhanced since the authors have stated that the anisotropic absorption is unchanged. Once the device is fabricated, the photoresponse PR is independent of the channel's anisotropy, but only depends on the anisotropic absorption. So, the statement like "As you noticed, the ferroelectric field does not change the dichroic absorption of BP. The PR enhancement of the device in Figure 4a is the result of the ZT being enhanced in the armchair direction" sounds contradictory.

c. I also don't understand why the PN junction could enhance PR. Such a statement cannot be found in the citations given by the authors. For example, the NP 2018 paper only showed the results and claimed that their PR is reasonable. The NN 2015 paper showed a PR of 3.5, which cannot be used to support the PR=288 reported here.

2. Presentation: I am convinced by the authors that DFT may not be necessary.

3. Noise: Nice efforts. Just one follow-up question: why the noise spectra are flat at low frequency? I thought $1/f$ noise should dominate there. No revisions in the manuscript are required.

4. Research background: Nice efforts. Since the authors have acknowledged it is difficult to acquire the complete polarization information of BP along 0° , 45° and 90° in one flake, I would suggest the authors add one or two sentences in the discussion, providing an outlook of the commercialization potential of this approach. For example, to achieve four orientations, maybe four steps of transferring (wafer-level BP) should work?

5. Photoresponse in Ferroelectrics. The authors have made a common mistake, that is, the bulk photovoltaic photocurrent is NOT necessary to be along with the ferroelectric polarization direction. Although this misconception is common, even including paper on Science. Please refer to Prof Marin Alexe's work: (1) "Bulk photovoltaic effect in monodomain BiFeO₃ thin films", APL, 2017; (2) "Role of domain walls in the abnormal photovoltaic effect in BiFeO₃", Nature Com, 2013.

6. Data from other refs. Nice efforts.

7. Abstract. No problem now.

8. PTE vs PV. Nice efforts. I would suggest the authors add proper references to support their claim of "This slit-shaped current mapping and non-linear output curve indicate it is a typical PN junction device, whose photeresponse (a typo in the manuscript) is dominated by PV effect".

Reviewer #3 (Remarks to the Author):

All my concerns have been well addressed by authors. This work can be published in current form.

Response to Reviewers' Comments

Reviewer 2:

Comments: Thanks for the authors' substantial efforts in answering my questions. While the manuscript has been greatly improved, I feel that some core concerns remain, and I cannot fully recommend the publication in its current form. Please see my comments below:

Answer: We would like to thank the reviewer for carefully reading the manuscript and providing meaningful comments. These comments are instructive and actually improve our manuscript. These issues have been addressed and answered one by one as follows:

1. The mechanism is still not clear to me.
 - a. The authors have used the enhancement of ZT to explain the observed anisotropic PTE, but this is very strange, if not irrelevant. Note that the enhancement of ZT values usually rely on the suppression of the thermal conductivity of lattice while maintaining that of electrons, and this is why engineering phonons is a common approach to increase ZT. Look at the definition of ZT, then we understand that the increase of ZT may not necessarily be due to the increase of Seebeck coefficient (S), and in fact usually not the case. Therefore, the authors should carefully check their analysis and avoid being illogical.

Answer: Thank for your valuable comment. We have carefully studied strategies to enhance the ZT of thermoelectric materials and made some corrections in the manuscript. The thermoelectric conversion efficiency (ZT) can be expressed as $ZT = S^2\sigma/K$, where S, σ and K stand for Seebeck coefficient, electrical conductivity and thermal conductivity, respectively. The S and σ are dominated by the electronic band structure.

The thermal conductivity of black phosphorous is anisotropic because of the anisotropic phonon dispersion. There are usually two main directions for obtaining low K, one is to enhance electron-phonon interactions through introducing scattering

centers, and the other is to use materials with intrinsically low thermal conductivity. Zhu et al. found that enhanced electron-phonon scattering helps reduce the thermal conductivity of heavily doped Si (Advanced Electronic Materials, 2016, 2, 1600171). In view of the polarization Raman spectrum, the ferroelectric field enhances electron-phonon interaction at 0° incident light, while it does not change at 90° incident light. So, the thermal conductivity along the armchair direction should be reduced and the ZT in the armchair direction should be improved.

For S and σ , as shown in Figure R2-1, Shimizu et al. reported that the ZT of BP can be controlled by using electric field to tune S and σ (Nano Letters, 2016, 16, 4819). Therefore, the ZT of n-doped and p-doped BP is different and the direction of PTE photocurrent should be same as PV photocurrent.

Figure R2-1. (a) Seebeck coefficient as a function of gate voltage. (b) Three-dimensional conductivity as a function of gate voltage. (Nano Letters, 2016, 16, 4819)

Essentially, S and σ are dominated by the electronic band structure, and electronic band engineering is the leading strategy to enhance ZT. As shown in Figure R2-2, the carrier mobility along the light-band direction is higher than that along the heavy-band structure, while the Seebeck coefficient remains similar. Zhu et al. utilized the valley anisotropy to enhance the ZT along the c-axis of Mg_3Sb_2 (Nature Communications, 2021, 12, 5408). Moreover, many researches have proved that the band structure of BP shows more obvious anisotropy under the modulation of a vertical electric field (Science, 2015, 349, 723; Nano Letter 2017, 17, 1970). Considering the anisotropic band structure, it is not difficult to infer that the ferroelectric field can enhance the carrier mobility along the armchair direction. Thus, the ZT ratio ($ZT_{\text{armchair}}/ZT_{\text{zigzag}}$)

also increases. We have made some corrections in our manuscript (Page 5, Line 108-110; Page 7, Line 154-155; Page 9, Line 210-214; Page 10, Line 248-253; Page 14, Line 360-362).

Figure R2-2. (a) The difference in the μ of light- and heavy-bands and along different directions of the anisotropic band. (b) Temperature dependences of ZT ratio along ab-plane and c-axis. (Nature Communications, 2021, 12, 5408)

b. Even we assume that the Seebeck coefficient can be indeed enhanced along the armchair direction, I still don't understand how the PR can be enhanced since the authors have stated that the anisotropic absorption is unchanged. Once the device is fabricated, the photoresponse PR is independent of the channel's anisotropy, but only depends on the anisotropic absorption. So, the statement like "As you noticed, the ferroelectric field does not change the dichroic absorption of BP. The PR enhancement of the device in Figure 4a is the result of the ZT being enhanced in the armchair direction" sounds contradictory.

Answer: Thank for your kind comment. Anisotropic absorption plays an important role in polarization detection. However, there is a complicated photoelectric conversion process. Many studies have reported that it is possible to manipulate the electric property and photocarrier generation, separation and transport processes by applying an external gate voltage (Nature Communications, 2019, 10, 2302; Nature Nanotechnology, 2013, 8, 952). Through polarization-dependent infrared spectroscopy measurements in Supplementary Fig. 2, we confirm that the ferroelectric field does not change the anisotropic absorption of BP. So, the ferroelectric field should change the

photoelectric conversion and carrier transport processes. ZT represents thermoelectric conversion efficiency, and a large ZT leads to a large photocurrent. As shown in Figure R2-3, the ZT along the armchair direction can be enhanced by the ferroelectric field. According to polarization Raman spectroscopy measurements, the electron-phonon interaction in BP gets enhanced by ferroelectric field at 0° incident light. Therefore, the ZT of the device under 0° incident light should be larger than that under 90° incident light, and the polarization photocurrent ratio gets improved.

Figure R2-3. Gate-tuned thermoelectric property in BP.

c. I also don't understand why the PN junction could enhance PR. Such a statement cannot be found in the citations given by the authors. For example, the NP 2018 paper only showed the results and claimed that their PR is reasonable. The NN 2015 paper showed a PR of 3.5, which cannot be used to support the PR=288 reported here.

Answer: Thanks for your nice comment. Many scientists have reported the phenomenon that polarization sensitivity can be enhanced by fabricating heterojunctions. In the NN 2015 paper (Nature Nanotechnology, 2015, 10, 707), the authors mentioned that the photocurrent with 0° polarization light can be enhanced by one order of magnitude by doping BP into PN junction, while the photocurrent with 90° polarization light does not change. The photocurrent enhancement originates from the PN junction tuned by gate voltage. As mentioned in the original article, "Namely, as V_G increases, the tunable perpendicular electric field can separate the electrons and holes-

electrons move on the surface and holes move in the bulk, which can reduce their recombination and increase the value of n_0 .” n_0 is the number of electron-hole pairs generated by the light shining on the sample and large n_0 leads to large photocurrent. In another paper (ACS Nano, 2019, 13, 9907), One reason of the enhanced polarization sensitivity is also the built-in electric field. As mentioned in the original article, “The built-in perpendicular electric field in the vertical hetero-junction can serve to spatially separate the photogenerated carriers, and reduce the recombination probability for electrons and holes during their transportation”.

On the other hand, enhanced PR maybe related to the shape of the junction. As shown in Figure R2-4, we doped MoTe₂ into PN homojunction by the same method, and the isotropic 2D material MoTe₂ shows a polarization photoresponse with PR about 1.32.

Figure R2-4. Polarization photoresponse in MoTe₂ homojunction.

2. Presentation: I am convinced by the authors that DFT may not be necessary.

Answer: Thank again for your comment.

3. Noise: Nice efforts. Just one follow-up question: why the noise spectra are flat at low frequency? I thought $1/f$ noise should dominate there. No revisions in the manuscript are required.

Answer: Thank you very much for your meaningful comment. As shown in Figure R2-5, the noise current spectrum of photovoltaic device is dominated by generation-recombination noise, which is frequency-independent at low frequencies. (Nature Photonics, 2018, 12, 601). The BP PN homojunction device works in photovoltaic

mode at zero bias. So, the shape of the measured noise current spectrum of device is common in systems dominated by generation-recombination noise. The $1/f$ noise is not observed at low frequencies.

Figure R2-5. Spectral noise density of a bP/MoS₂ photodiode. (Nature Photonics, 2018, 12, 601)

4. Research background: Nice efforts. Since the authors have acknowledged it is difficult to acquire the complete polarization information of BP along 0°, 45° and 90° in one flake, I would suggest the authors add one or two sentences in the discussion, providing an outlook of the commercialization potential of this approach. For example, to achieve four orientations, maybe four steps of transferring (wafer-level BP) should work?

Answer: Thank you for valuable suggestion. Polarization imaging collects information about objects in complicated environment. To obtain full polarization information, a division of focal plane (DoFP) polarization camera integrates a micro-polarizer array with four polarization direction (0°, 45°, 90° and 135°) units into the super-pixels of the focal plane array sensor (Optics Express, 2021, 29, 22066). The structure of DoFP can be a reference if using the method in our manuscript to design polarization imaging system. So, we can transfer the BP flakes and make sure these BP flakes along four crystal orientation when fabricating devices. To providing an outlook on commercialization potential, we have added some sentences in this manuscript (Page 14, Line 372-374).

5. Photoresponse in Ferroelectrics. The authors have made a common mistake, that is,

the bulk photovoltaic photocurrent is NOT necessary to be along with the ferroelectric polarization direction. Although this misconception is common, even including paper on Science. Please refer to Prof Marin Alexe's work: (1) "Bulk photovoltaic effect in monodomain BiFeO₃ thin films", APL, 2017; (2) "Role of domain walls in the abnormal photovoltaic effect in BiFeO₃", Nature Com, 2013.

Answer: Thank you for the friendly suggestion. According to Prof Marin Alexe's works (Applied Physics Letters, 2017, 110(18): 183902, Nature Communications, 2013, 4, 2835), the bulk photovoltaic effect originates from the non-centrosymmetry of ferroelectric semiconductors. The direction of bulk photovoltaic current in BiFeO₃ film should depend on the light polarization direction and the working temperature, which is not along the ferroelectric polarization direction. In our work, the organic ferroelectric film P(VDF-TrFE) is an insulator and have no photoresponse to visible and near-infrared light. Therefore, the bulk photovoltaic effect can be neglected.

6. Data from other refs. Nice efforts.

Answer: Thanks again for your careful reading.

7. Abstract. No problem now.

Answer: Thank you very much for your valuable comment.

8. PTE vs PV. Nice efforts. I would suggest the authors add proper references to support their claim of "This slit-shaped current mapping and non-linear output curve indicate it is a typical PN junction device, whose photeresponse (a typo in the manuscript) is dominated by PV effect".

Answer: Thanks for your useful suggestion. We have added some references (Nature Communications, 2019, 10, 3331; Advanced Materials, 2019, 31, 1902044) to the manuscript and corrected the typo in the manuscript (Page 12, Line 311-312). As described in the original article, "In PV detectors, the photoresponse is localized around the junction interface, while the spatial profile of photocurrent in PTE detectors can be quite broad." (Advanced Materials, 2019, 31, 1902044). Moreover, we fabricated MoTe₂ PN homojunction devices via the same method in our manuscript, and the optical image and scanning photocurrent mapping spectroscopy are shown in Figure R2-6. The photoelectric conversion process occurs at the junction region defined by

ferroelectric domains.

Figure R2-6. Optical image and photocurrent mapping of MoTe₂ PN junction. The inset is the PFM phase.

REVIEWERS' COMMENTS

Reviewer #2 (Remarks to the Author):

The authors have addressed all my concerns, and I recommend the publication as it is. Congratulations to the authors for their great efforts.